# Experimental determination of excitonic band structures of single-walled carbon nanotubes using circular dichroism spectra

Xiaojun Wei[1], Takeshi Tanaka[1], Yohei Yomogida[1], Naomichi Sato[2], Riichiro Saito[2] & Hiromichi Kataura[1]

Experimental band structure analyses of single-walled carbon nanotubes have not yet been reported, to the best of our knowledge, except for a limited number of reports using scanning tunnelling spectroscopy. Here we demonstrate the experimental determination of the excitonic band structures of single-chirality single-walled carbon nanotubes using their circular dichroism spectra. In this analysis, we use gel column chromatography combining overloading selective adsorption with stepwise elution to separate 12 different single-chirality enantiomers. Our samples show higher circular dichroism intensities than the highest values reported in previous works, indicating their high enantiomeric purity. Excitonic band structure analysis is performed by assigning all observed $E_{ii}$ and $E_{ij}$ optical transitions in the circular dichroism spectra. The results reproduce the asymmetric structures of the valence and conduction bands predicted by density functional theory. Finally, we demonstrate that an extended empirical formula can estimate $E_{ij}$ optical transition energies for any ($n,m$) species.

[1] Nanomaterials Research Institute, National Institute of Advanced Industrial Science and Technology (AIST), Tsukuba 305-8565, Japan. [2] Department of Physics, Tohoku University, Sendai 980-8578, Japan. Correspondence and requests for materials should be addressed to H.K. (email: h-kataura@aist.go.jp).

For more than 20 years since the discovery of single-walled carbon nanotubes (SWCNTs), we have lacked accurate knowledge of their unique optical and physical properties due to variations in the allowed structures (n,m), which have been defined using chiral indices[1]. It is known that $E_{ii}$ optical transitions are allowed for the polarization of the electric field of light in parallel to the nanotube axis, whereas $E_{ij}$ ($i_v \rightarrow j_c$) transitions are allowed for cross polarization[2–5]. Usually, only the $E_{ii}$ transitions can be observed from optical absorption and photoluminescence spectra because a strong local field induced by the applied field obscures the cross-polarized optical transitions[3–5]. Limited information about the $E_{ii}$ energies alone is insufficient for constructing valence and conduction band structures that require $E_{ij}$ values. Although Miyauchi et al[3,4]. succeeded in measuring the cross-polarized optical transitions using aligned SWCNTs, the results were limited. Previously, the detailed band structures of specific SWCNTs could only be obtained by density functional theory (DFT) calculations as predictions[6] except in the case of experimental studies utilizing scanning tunnelling spectroscopy[7,8]. On the other hand, the DFT calculations suggest that all optically allowed transitions $E_{ii}$ and $E_{ij}$ are observable in circular dichroism (CD) spectra, even for randomly aligned chiral SWCNTs[9–11] due to the comparable signal intensities of both transitions. Although SWCNTs have one-dimensional electronic systems, an interaction between right- or left-handed SWCNTs and circularly polarized light should exist. The values of all $E_{ii}$ and $E_{ij}$ transitions allow us to analyse the detailed band structures of the SWCNTs with more precision than can be implemented by other optical measurements.

A mixed chirality sample cannot be used for the band structure analysis because numerous peaks are expected in the CD spectra. Furthermore, this analysis requires the preparation of not only high-purity single-chirality but also highly selected enantiomers because the CD intensity is proportional to the differential ratios between left- and right-handed SWCNTs. Peng et al.[12] reported the first enantiomer separation using a specially designed chiral dispersant molecule with different affinities for the left- and right-handed enantiomers[13], and Ghosh et al.[10] used a nonlinear density-gradient ultracentrifugation (DGU) to achieve enantiomer separation using a common chiral dispersant. However, the purities of these samples were not high enough for the band structure analysis because the observed CD peaks were difficult to assign to a specific chirality. Gel column chromatography with the overloading technique has also been used for the enantiomer separation, but the enantiomer purity was not high enough[14]. For the detailed analysis of the band structure, even very weak peaks must be recognized. Higher enantiomer purity is required for obtaining a high signal/noise ratio in a CD spectrum.

To improve both single-chirality purity and enantiomer purity, we developed a separation method that combines selective adsorption with the overloading technique and selective desorption[15] with a stepwise elution gel chromatography. Using a dextran-based gel for the column medium and a mixed surfactant for the eluent, this method can simultaneously separate single-chirality SWCNTs and enantiomers based on the effects of the optically active dextran. Using this technique, here we successfully separate 12 distinct (n,m) single-chirality species of semiconducting SWCNTs and measure their CD spectra. The CD intensity, which is normalized by its optical absorbance, is higher than that of previous reports to the best of our knowledge, indicating that the highest enantiomer purity is obtained[10,13,14]. Excitonic band structure analysis is also performed by assigning all the CD peaks to allowed optical transitions. Our results are in excellent agreement with the DFT calculations, which include asymmetries of the valence and conduction bands[6]. Interestingly, a plot of the allowed $E_{ij}$ transitions against the SWCNT diameters

shows a typical 'family behaviour', which suggest that our results are quite reasonable. Our challenge of excitonic band structure analysis is based on a simple analysis of the CD spectrum, but it originates from an interaction between a circular polarized photon and the chiral structure of the carbon network. The results presented here could be an important reference for further studies on the physical properties of SWCNTs. Moreover, the new separation method developed in this work may be important for possible biochemical and medical applications because the high-purity SWCNT enantiomers could be used for chiral separation and the synthesis of chiral molecules[16].

## Results

**Single-chirality enantiomer separation and evaluation.** In this work, higher-purity separation was achieved by combining the overloading technique for the C–C bond curvature-selective adsorption of semiconducting SWCNTs[17] with diameter-selective desorption by stepwise elution chromatography[15]. In the first separation step, 80 ml of a SWCNT dispersion in 3% SDS solution was injected onto a multicolumn filled with 5.6 ml of the dextran-based gel to adsorb semiconducting SWCNTs with very narrow C–C bond curvature distributions. After sufficiently flushing the column, 1% sodium cholate (SC) solution was injected for elution. From the first separation step, we obtained a mixture of several species of SWCNTs in a 1% SC solution (Supplementary Fig. 1). Before the second separation step, the same amount of 1% SDS solution was added to the SWCNTs solution to create a 0.5% SC + 0.5% SDS mixed solution. This SWCNT solution was injected onto the gel column, and then stepwise elution chromatography was performed using an aqueous solution of 0.5% SC + 0.5% SDS + x% sodium deoxycholate (DOC). For the diameter-selective elution, the DOC concentration x was increased stepwise from 0.01 to 0.07%. In this process, single-chirality species with smaller diameters were eluted with lower DOC concentrations. The relationship between the separation order of (n,m) nanotubes and DOC concentration is consistent with our previous report of diameter-selective desorption[15,18] (Supplementary Fig 2–4). Importantly, each chirality eluted at two different DOC concentrations, $x_1$ and $x_2$ (for example, (6,5) SWCNTs were eluted at 0.030 and 0.035% DOC, Supplementary Fig. 2d), which corresponded to two enantiomers with the same band structure but different affinities to the gel. Finally, we measured their CD spectra and confirmed enantiomer separation (( + )- or ( − )-(n,m), described in detail below). This separation method of stepwise elution chromatography is effective not only for single-chirality separation but also simultaneous enantiomer separation. Compared with our previous work[14], a higher resolution of single-chirality enantiomeric separation was achieved in this study, which is mainly contributed by the introduced mixed-surfactant system and the stepwise elution procedure[15].

Figure 1a shows the optical absorption spectra of the 12 single-chirality enantiomers of semiconducting SWCNTs obtained in this work. The spectra of either ( + )- or ( − )-(n,m) enantiomers with higher purities are shown here. Each spectrum exhibits a series of sharp absorption peaks corresponding to the $E_{11}$, $E_{22}$, $E_{33}$ and $E_{44}$ transitions with very low baseline absorption, which is known as a typical feature of chirality-sorted SWCNTs. These species were identified as (6,4), (9,1), (8,3), (6,5) (7,3), (7,5), (10,2), (9,4), (8,4), (7,6), (10,3) and (11,1) from their $E_{11}$ and $E_{22}$ peak positions[19,20]. Notably, the baseline absorption of each sample was very low, indicating that impurities such as nanotube aggregates, amorphous carbon, and catalyst particles, were successfully removed in the separation processes[21,22]. The purity of each (n,m) species was basically calculated as an

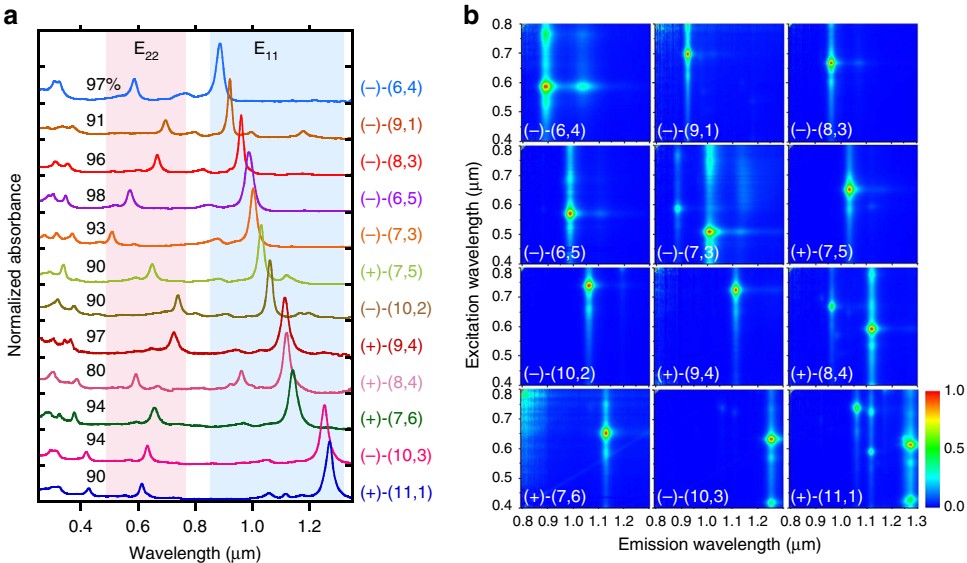

**Figure 1 | Optical evaluation of the separated 12 (n,m) species. (a,b)** Optical absorption spectra (**a**) and photoluminescence contour maps (**b**) of the 12 sorted (n,m) semiconducting SWCNTs.

integrated intensity ratio of the absorption peak of the target chirality to the sum of all other peaks using PeakFit software (Supplementary Fig. 5). Since the absorption cross section of each (n,m) species is different[23,24], we have corrected the above purity using a theoretical formula of chirality-dependent oscillator strengths reported in ref. 23. The final purity of each species is indicated in Fig. 1a. Most of the samples were more than 90% pure except for species (8,4). Because the band structure analysis requires high-purity, we selected only high-purity species to maintain the reliability of the analysis. The purities of all species except (8,4) were the same or higher than those of previous reports, including studies using DGU[10], DNA wrapping[25] and gel chromatography[17,26] (Supplementary Table 1). Figure 1b shows the photoluminescence (PL) excitation-emission mapping of 12 sorted species corresponding to Fig. 1a. In general, because the $E_{22}$ excitation - $E_{11}$ emission peak is dominant in the PL maps, each map had only one bright peak. In addition, the phonon sideband peak is well known to be observed in the excitation spectrum for the $E_{11}$ emission. In the cases of (10,3) and (11,1), the $E_{11}$ emission by $E_{33}$ excitation ($\sim$420 nm) was also observed. Furthermore, we sometimes observe a PL peak from oxide states that were unexpectedly produced in the ultrasonic dispersion process[27–29]. Because the PL intensity of the oxide state was much higher than the original peak, this peak was noticeably bright. Natural oxidation is diameter-dependent and is more frequently observed in small diameter SWCNTs with highly curved side walls, such as (6,4) at 1,050 nm. In general, most of the peaks on the PL map could be assigned to the original $E_{11}$ transitions, phonon side bands, and oxide states, indicating that high-purity single-chirality SWCNTs were present. In the PL map of (11,1), however, four additional peaks from (10,2), (10,3), (9,4) and (8,4) were observed, while very weak peaks were also observed in the absorption spectra. This suggested that the virtual PL efficiency of (11,1) was lower than that of the impurities, which was probably due to a self-absorption. After obtaining a high-purity single-chirality sample, we could estimate the PL efficiency by measuring the PL intensity, but this was not performed in this work. We assumed that the purity should not be determined by PL intensity, but instead by optical absorption.

Figure 2 shows the CD spectra and optical absorption spectra for the 12 different (n,m) enantiomers. In the CD spectra, the SWCNTs that exhibited a positive $E_{22}$ CD peak were denoted

as (−), whereas the SWCNTs that exhibited a negative $E_{22}$ CD peak were denoted as (+)[10,30]. Here, (+) and (−) refer to enantiomers that showed plus and minus $E_{11}$ CD peaks, respectively. Both (+)- and (−)-(n,m) were obtained for all types except (9,4) and (10,3). The missing (9,4) and (10,3) enantiomers were mixed with the other chiralities and could not be recovered as a single chirality. By comparing the CD spectra with the optical absorptions, the CD peak positions of $E_{22}$, $E_{33}$ and $E_{44}$ were determined and were used for the assignments. A photograph of the separated single-chirality SWCNT enantiomers is shown in Supplementary Fig. 6, in which different colours for each chirality indicate different electronic structures; (+)- and (−)-enantiomers are shown in the same colour when the concentrations are same. For example, the same concentrations of (+)- and (−)-(6,5) enantiomers are shown in exactly the same colour (Supplementary Fig. 7).

The relative enantiomeric purity of sorted enantiomers can be evaluated by a normalized CD intensity ($CD_{norm}$)[13,14], instead of commonly used anisotropic factor that cannot be defined for SWCNT. Table 1 summarizes the $CD_{norm}$ values of individual (n,m) enantiomers obtained in this work and in previous reports. All $CD_{norm}$ values obtained in this study were higher than those obtained using the repeated overloading method[14], temperature controlled gel chromatography[14], DGU[10] and molecular recognition[13], indicating that the highest enantiomeric purity was obtained in the present study. The conventional enantiomeric purity of (6,5) enantiomers was roughly estimated to be higher than 90% for both (+)- and (−)-enantiomers. A detailed analysis should be discussed elsewhere.

**Alternative behaviour of CD between Type 1 and Type 2.** The order of the affinity between the SWCNT enantiomers and the dextran-based gel was scaled by the corresponding DOC concentrations of the stepwise elutions. Figure 3 shows the results of the stepwise elution for the 12 different single-chirality SWCNT enantiomers as a function of nanotube diameter. A clear relationship between the diameter and the DOC concentration can be seen; namely, the required DOC concentration was higher for SWCNTs with larger diameters. Interestingly, for the separation of Type 1 enantiomers ($|n-m| = 3a + 1$, where 'a' is an integer[19,20,31]; (7,3), (6,5) (7,6) and (8,4)), higher DOC

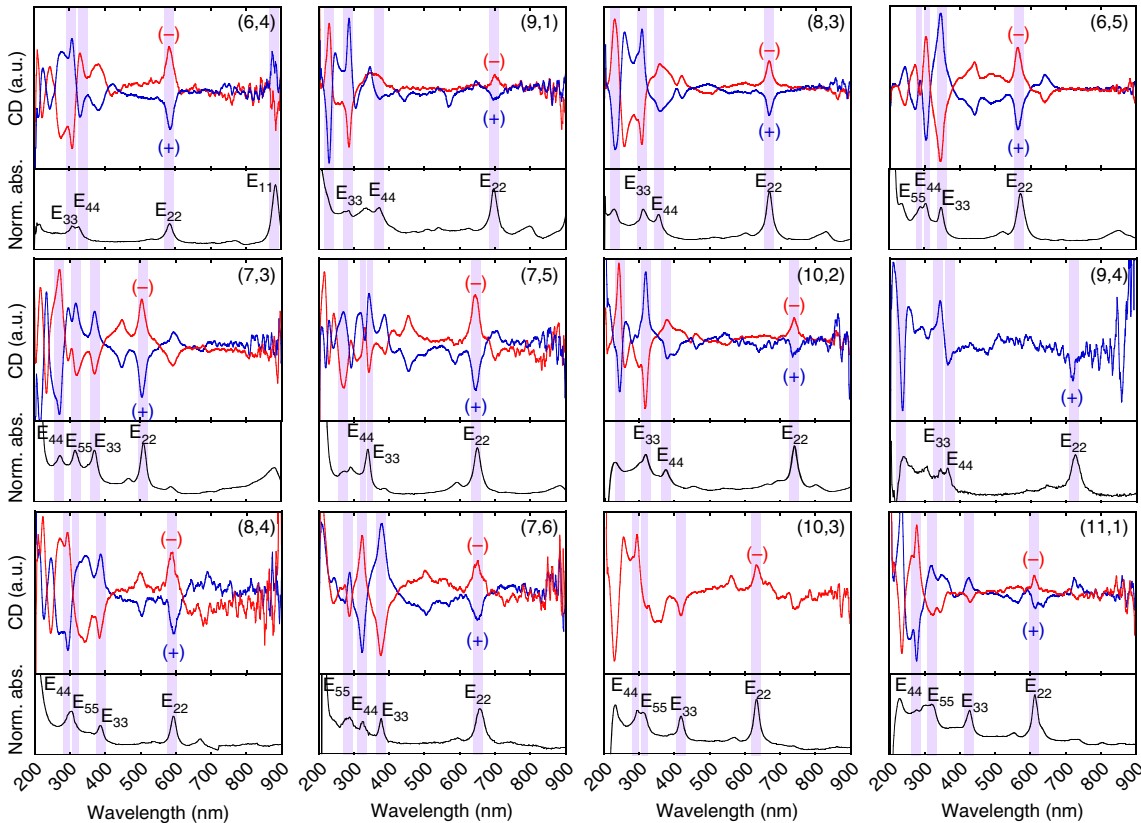

**Figure 2 | CD spectra of the separated single-chirality enantiomers.** CD spectra and optical absorption spectra of the 12 different (*n,m*) species. The CD intensities were normalized for all 12 samples.

**Table 1 | CD_norm* values of sorted single-chirality SWCNT enantiomers.**

| (*n,m*) | (6,4) | | (9,1) | | (7,3) | | (6,5) | | (8,3) | | (7,5) | | (8,4) | | (7,6) | | (9,4) | | (8,6) | | (8,7) | | (10,2) | | (10,3) | | (11,1) | |
|---|---|---|---|---|---|---|---|---|---|---|---|---|---|---|---|---|---|---|---|---|---|---|---|---|---|---|---|---|
| Enantiomers | (−) | (+) | (−) | (+) | (−) | (+) | (−) | (+) | (−) | (+) | (−) | (+) | (−) | (+) | (−) | (+) | (−) | (+) | (−) | (+) | (−) | (+) | (−) | (+) | (−) | (+) | (−) | (+) |
| This work | 61 | 62 | 21 | 13 | 56 | 55 | 85 | 93 | 50 | 41 | 61 | 27 | 81 | 87 | 42 | 73 | - | 81 | - | - | - | - | 27 | 22 | 61 | - | 19 | 12 |
| Gel: repeated overloading[14] | 7 | - | - | - | - | <1 | 2 | 10 | - | - | 6 | 4 | 4 | <1 | 1 | 2 | 12 | 4 | - | 4 | <1 | - | - | - | - | - | - | - |
| Gel: temp. control[14] | 11 | - | - | - | - | - | 7 | - | - | - | 7 | 2 | 6 | - | - | 6 | - | - | 4 | - | - | - | - | - | - | - | - | - |
| DGU[10] | 58 | 40 | - | - | 37 | 45 | 30 | 31 | 24 | 27 | - | 23 | 23 | 22 | - | - | - | - | - | - | - | - | - | - | - | - | - | - |
| Molecular recognition[13] | - | - | - | - | - | - | 24 | 24 | - | - | - | - | - | - | - | - | - | - | - | - | - | - | - | - | - | - | - | - |

CD, circular dichroism; SWCNT, single-walled carbon nanotube.
*CD_norm (mdeg) was defined as $CD_{norm} = (CD_{raw} / L_{CD}) / (A_{E_{22}} / L_{abs})$, where $CD_{raw}$ is the CD intensity at the $E_{22}$ transition, $A_{E_{22}}$ is the absorbance at the $E_{22}$ transition, and $L_{CD}$ and $L_{abs}$ are the path lengths of the optical cuvettes used for the measurements (both 10 mm in this study)[13,14].

concentrations were required for the (−)-enantiomers than that for the (+)-enantiomers, whereas lower DOC concentrations were required for the (−)-enantiomers than for the (+)-enantiomers for Type 2 (|*n* − *m*| = 3*a* + 2; (6,4), (9,1), (8,3) and (7,5)). A similar relationship was obtained in the separation of the (10,2), (11,1) and (10,3) enantiomers, which are large diameter enantiomers requiring the use of 0.5% SC + 1% SDS as a solvent instead of 0.5% SC + 0.5% SDS (Fig. 3, inset). The enantiomer separation was probably achieved due to the interaction between the optically active SWCNT and the optically active dextran-based gel. If this assumption is correct, the separation order of handedness should be the same for all single-chirality SWCNTs. However, our results showed that the separation order of (+)- and (−)-enantiomers depended on the type of SWCNT index. A possible explanation is that the CD signals of Type 1 and Type 2 SWCNTs with a specific handedness have opposite signs. This statement is supported by theoretical calculation of CD spectra of SWCNTs described below.

In Fig. 4a, we show the calculated CD signal ΔW as a function of the wavelength of the incident circular polarized lights in the direction perpendicular to the nanotube axis, for (6,5) (red line) and (11,−5) (blue line) SWCNTs in which (*n,m*) and (*n + m,* − *m*) SWCNTs are enantiomers to each other. In the numerical calculation, ΔW is defined by the difference of optical absorption probability of (6,5) (or (11,−5)) SWCNT for the left- and right-handed circular polarized lights within tight-binding method for optical transition of SWCNTs[32]. It is important to note that the optical absorption probability near the K point for the right-handed circular polarized light is identical to that near the K′ point for the left-handed circular polarized light because of the time reversal symmetry between the K and K′ points, if we adopt an approximation of the vertical optical transition in the *k* space. It means that we would get ΔW = 0 if we adopted this approximation. Thus it is essential in the calculation of ΔW to consider the phase difference of the light for different carbon atom positions that is usually neglected in the calculation of the absolute values of the optical

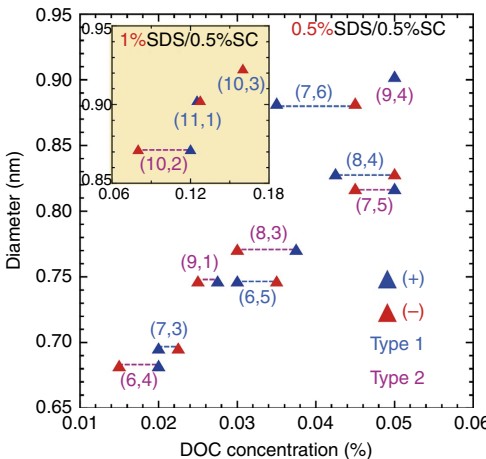

**Figure 3 | Relationship between DOC concentration and diameter.** The SWCNTs eluted at higher DOC concentrations have higher affinity toward the dextran-based gel. Inset: the relationships for (10,2), (11,1) and (10,3) SWCNTs separated in the solvent containing 0.5% SC and 1% SDS (see Supplementary Methods and Supplementary Fig. 4). Blue and red triangles denote ( + )- and (−)-SWCNTs, respectively. Type 1 and Type 2 chirality (n,m) are denoted as blue and pink letters, respectively.

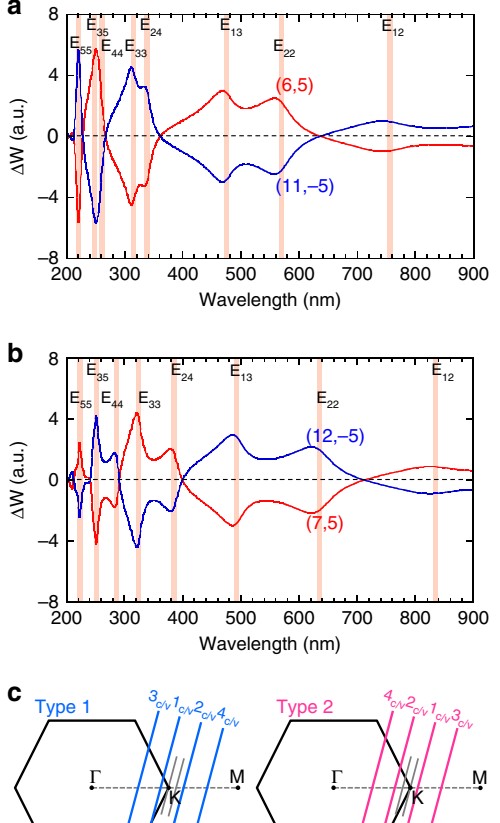

**Figure 4 | Theoretical calculation of CD spectra of enantiomers.** (**a,b**) Calculated CD signal ΔW of enantiomers for (**a**) (6,5) (red) and (11, − 5) (blue) and (**b**) (7,5) (red) and (12, − 5) (blue), as a function of wavelength of incident light. Shaded lines correspond to E$_{ii}$ and E$_{ij}$ van Hove singularity of joint density of states. (**c**) Reciprocal space of the graphene and cutting lines corresponding to Type 1 (left of panel) and Type 2 (right of panel) SWCNTs.

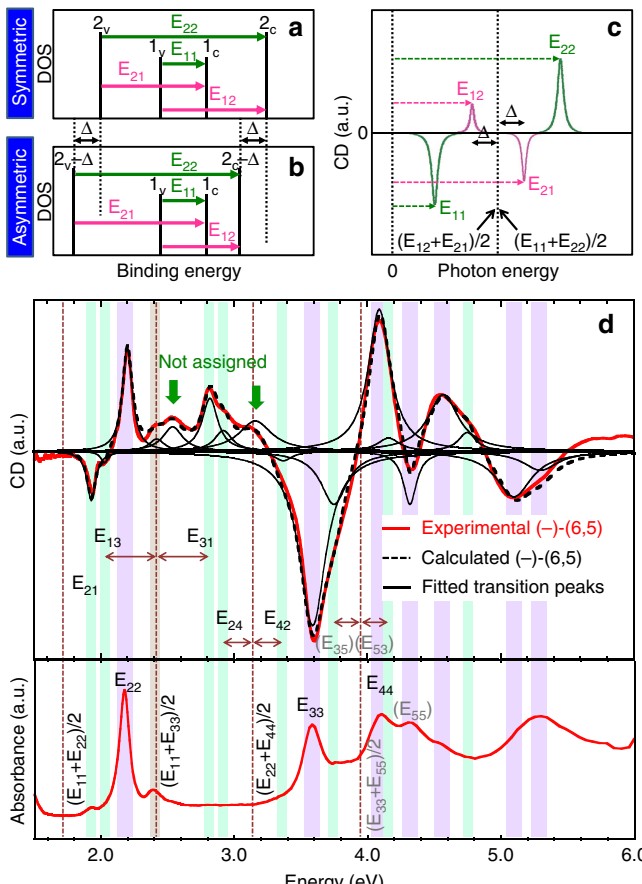

**Figure 5 | Analysis of CD and optical absorption spectra.** (**a**) Schematic diagrams of a symmetric band structure. (**b**) Schematic diagrams of an asymmetric band structure with a shift of Δ for 2$_c$ and 2$_v$ bands compared with **a**. (**c**) Schematic CD spectrum of a ( − )-enantiomer with the asymmetric band structure shown in **b**. (**d**) CD spectra (upper panel) and absorption spectra (lower panel) of the sorted ( − )-(6,5) SWCNTs. Four vertical dashed lines indicate the center positions of E$_{12}$/E$_{21}$, E$_{13}$/E$_{31}$, E$_{24}$/E$_{42}$, and E$_{35}$/E$_{53}$. The E$_{ii}$ and E$_{ij}$ transitions are depicted as purple and green strips, respectively. Individual single Lorenzian peaks were plotted using the energy values of all assigned transitions. The calculated CD spectrum (black dashed curve) is represented as the sum of all plotted peaks for comparing the experimental spectrum (red solid curve). Two unassigned peaks are labelled by green arrows.

absorption. The detail of the calculation will be reported elsewhere.

The calculated results reproduce the experimental results in many aspects: (1) ΔW of (6,5) SWCNT changes the sign at E$_{22}$, E$_{33}$, and E$_{44}$ and (2) ΔW of (11, − 5) SWCNTs gives the opposite sign to that of (6,5) for all the wavelength of the light. Further, when we change (n,m) from (6,5) (Type 1 SWCNT) to (7,5) (Type 2 SWCNT) as shown in Fig. 4b, we get the opposite sign of ΔW to that for (6,5) for each E$_{ii}$ energy. This phenomenon can be explained by the difference in the reciprocal space of the graphene and cutting lines between Type 1 and Type 2 SWCNTs as shown in Fig. 4c. The optical absorption for E$_{ii}$ van Hove singularity of Type 1 SWCNT occurs at the opposite k points measured from the K point compared with that of Type 2 SWCNT, whose transition optical dipole moment around the K point gives the opposite sign[33]. The opposite behaviour between Type 1 and 2 SWCNTs in the calculation is also consistent with the experimental expectation based on their type-dependent affinities with gel. Thus, alternative behaviour of ΔW as a

function of the wavelength comes from the change of the wavefunctions on one-dimensional cutting line near the K point[34,35].

It should be mentioned that we measure CD of SWCNTs with random orientation in the experiment. For incident circular polarized light, there are two inequivalent geometries of SWCNTs: The propagating directions of the light are (1) perpendicular to the nanotube axis (the case of Fig. 4a,b) and (2) parallel to the nanotube axis (not shown). Even though we should integrate the CD for possible direction of the light, we did not show this for simplicity. We can imagine that the case of (2) gives a minor contribution after integration of azimuth angle of spherical coordinate. Further, we did not consider the effect of the exciton in this calculation in the first approximation. Since the wavefunction of an exciton is given by linear combination of the Bloch functions along one-dimensional Brillouin zone (or cutting line)[36,37], we can say that the behaviours of CD do not change even after we consider the exciton effect. Thus the present calculated results are reasonable to simulate the experimental results.

**Asymmetric band structure analysis.** For the analysis of the CD peaks, we needed to know the selection rules for the CD signal and the related optical transitions for an individual SWCNT[38]. The energy bands and the allowed optical transitions of a semiconducting SWCNT are shown in Supplementary Fig. 8. Optical transitions from the valence band (denoted as $n_v$) to the conduction band (denoted as $n_c$) in a semiconducting SWCNT are characterized by $\Delta n$, where $\Delta n = n_c - n_v$. When the electric field polarization of incident light is parallel to the SWCNT axis, optical transitions with $\Delta n = 0$ ($E_{11}$, $E_{22}$, $E_{33}$...) are allowed, and when the polarization is perpendicular to the axis, optical transitions with $\Delta n = \pm 1$ ($E_{12}/E_{21}$, $E_{13}/E_{31}$, $E_{24}/E_{42}$...) are allowed[5,38]. Usually, only $E_{ii}$ transitions can be clearly observed in optical absorption spectra for randomly aligned SWCNTs because the intensity of the $E_{ii}$ transitions is much higher than the intensity of the $E_{ij}$ transitions. However, it is known that both $E_{ii}$ and $E_{ij}$ can be observed comparably in the CD spectra. Furthermore, based on this alternating behaviour of $E_{ii}$ CD signals of the ( + )- and ( − )-enantiomers, the $E_{11} \sim E_{55}$ signals of ( + )-enantiomers exhibit a ( +, −, +, −, + ) progression, whereas the signals of the ( − )-enantiomers exhibit the inverted ( −, +, −, +, − ) progression. This rule helped us to assign the deep $E_{ii}$ transitions, in which the trigonal warping effects tended to modify the $E_{ii}$ energy orders[9,10,39]. Actually, it has been reported that (6,4) has a higher $E_{33}$ value compared with its $E_{44}$ value[9,39]. Optical absorption cannot detect this crossover, but the sign of the CD signal can. Similarly, in this work, three $E_{ii}$ transition peaks were observed in the absorption spectrum of (8,3) SWCNTs, where the corresponding CD sign of the (–)-enantiomer exhibited a ( +, +, – ) progression from long to short wavelengths, as shown in Fig. 2. On the basis of the signs of the CD signals, the first peak at 668 nm could be assigned to the $E_{22}$ transition, and the following two peaks at 357 and 307 nm could be assigned to the $E_{44}$ and $E_{33}$ transitions, respectively. This crossover is consistent with the reported family pattern of higher order transitions[39].

Because a CD spectrum is equally sensitive to optical transitions of parallel and perpendicular polarizations, we could deeply analyse the band structure of SWCNTs using CD spectra. Figure 5a illustrates a symmetric band structure, in which $E_{21} = E_{12}$, and Fig. 5b shows a more realistic asymmetric band structure model. For both band structures, the values of $E_{11}$ and $E_{22}$ were the same, but the $2_v$ and $2_c$ bands were shifted relatively deeper than the $\Delta$ value for the asymmetric band (Fig. 5b). Using this simple asymmetric band structure model, we analysed an asymmetric SWCNT band structure for which $E_{21} - E_{12} = 2\Delta$.

For both the symmetric and asymmetric band structures, the same relationship of $(E_{11} + E_{22})/2 = (E_{12} + E_{21})/2$ was satisfied, meaning that the centre position of the $E_{11}$ and $E_{22}$ peaks were the same as the centre of the $E_{12}$ and $E_{21}$ peaks. This result is shown in a schematic CD spectrum in Fig. 5c. In this spectrum, the $E_{12}$ and $E_{21}$ transition peaks are located at the lower (left of panel) and higher (right of panel) energy sides of the centre of $E_{11}$ and $E_{22}$ by a distance of $\Delta$. Using this model, we found $E_{12}$ and $E_{21}$ peaks between the $E_{11}$ and $E_{22}$ peaks that were the same distance from the centre of the $E_{11}$ and $E_{22}$ peaks and had opposite signs[9]. Similarly, we found $E_{13}$ and $E_{31}$ between $E_{11}$ and $E_{33}$, $E_{24}$ and $E_{42}$ between $E_{22}$ and $E_{44}$, and so on. Here we did not consider the exciton binding energy difference in these optical transitions.

Figure 5d shows the CD spectra (upper panel) and absorption spectra (lower panel) of the ( − )-(6,5) SWCNTs separated in this work. Highly symmetric CD spectra of (6,5) enantiomers (shown in Fig. 2) indicated that our sample purity is high enough and we did not need to consider any impurities that might have affected the CD spectrum. All CD peaks could be assigned to (6,5) SWCNTs. First, we assigned $E_{ii}$ transitions by comparing the CD spectra with the absorption spectra, which are depicted as purple strips. After the determination of the $E_{ii}$ transition, the centre positions of $E_{ij}$ transition pairs ($E_{12}/E_{21}$, $E_{13}/E_{31}$, $E_{24}/E_{42}$ and $E_{35}/E_{53}$) were determined and depicted as vertical dashed lines. Then, we looked for $E_{ij}$ transitions on both sides of the centrelines at the same distance and found an appropriate set of $E_{ij}$ transitions, which are depicted as green strips in Fig. 5d. Because the sensitivity of the detector in the low energy region ($\sim 1.7$ eV) was not high enough to see the detail of the spectrum structure, we could not find the $E_{12}$ peak. Thus, the assigned $E_{21}$ peak did not have a partner $E_{12}$ peak. Although the transitions higher than $E_{44}$ can be observed in high energy region, it is difficult to accurately assign them due to the complicated CD peaks. Using all assigned peak energies, the CD spectrum was reproduced by Lorentzian oscillator fitting (black solid curves). The calculated CD spectrum (black dashed curve) was highly consistent with the experimental spectrum (red solid curve), except for two unassigned CD peaks at 2.54 eV, 3.14 eV. These peaks were probably contributed by phonon-assisted excitonic recombination, which was resolved in the PL measurement[40]. Under the same analysis rules, the CD peaks of the other 11 (n,m) species were assigned (Supplementary Fig. 9).

Using all the $E_{ii}$ and $E_{ij}$ transition energies, the valence and conduction band energy levels ($n_v$ and $n_c$) were determined for the 12 different sorted (n,m) species. The experimental results based on CD spectra were compared with the electronic density of states (DOS) calculated by DFT using local density approximation (LDA)[6] (see Supplementary Fig. 10). Generally, GW approximation gives more precise band structure including bandgap[41]. However, GW calculations were limited for very small diameter SWCNTs to date due to required huge resources. On the other hand, Saito et al. have already published DFT band structures for most of SWCNTs using LDA[6]. They also compared LDA and GW calculations for (7,0) SWCNT and found that well known bandgap underestimation in LDA can be corrected by simple bandgap enlargement[42]. Therefore, here we corrected DFT results by shifting $n_c$ and $n_v$ away from the Fermi level for + 0.15 and − 0.15 eV, respectively. On first observation, all SWCNTs appear to have asymmetric valence and conduction bands, which is roughly consistent with the DFT calculation. Since we modified LDA bandgap to fit the excitonic band gap, it is trivial that $1_v$ and $1_c$ levels show good agreement for most of species. Especially, the species (6,4), (8,3), and (10,2) showed good agreement in the second conduction and valence bands ($2_c$ and $2_v$). However, (6,5), (7,5), (9,4), (8,4), (7,6), (10,3) and (11,1) showed redshift of both

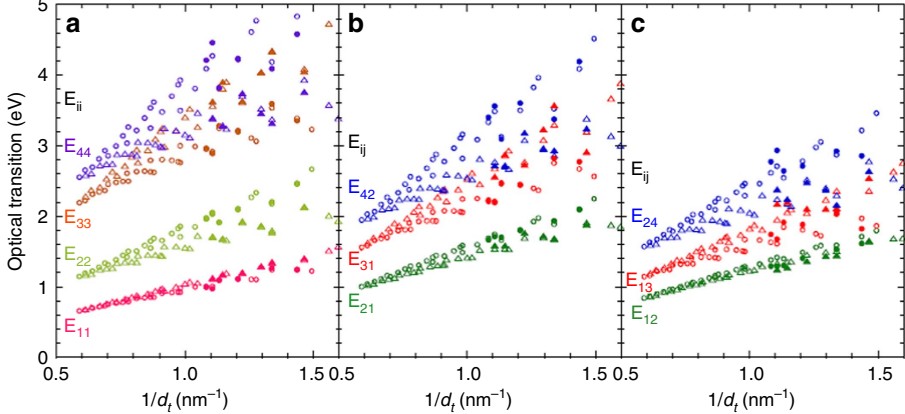

**Figure 6 | $E_{ii}$ and $E_{ij}$ transition energies.** (**a–c**) Transition energy of $E_{ii}$ and $E_{ij}$ as a function of $1/d_t$. Open and solid symbols (circles for Type 1 and triangles for Type 2 species) indicate the calculation and experimental results, respectively.

$2_c$ and $2_v$ bands keeping good agreement for $E_{22}$. Since $E_{22}$ and $E_{21}$ CD peaks were used to estimate asymmetric $2_v$ and $2_c$ levels, the redshift in $2_v$ and $2_c$ levels from LDA band is probably caused by the difference in exciton binding energy between $E_{21}$ and $E_{22}$ transitions. On the other hand, the species (9,1), and (7,3) disagreed with $E_{22}$ value from DFT calculation. This suggests that the exciton binding energies of $E_{11}$ and $E_{22}$ are different in these species. Most of species showed rather large deviations in the higher order energy bands ($n_c$, $n_v \geq 3$) probably due to more complicated excitonic and environmental effects. As we mentioned above, we did not consider the exciton effects in our analysis. This means that the band structures obtained in our analysis included the excitonic effects because all optical transitions in carbon nanotubes are modified by the one-dimensional exciton effect. By contrast, the LDA calculations did not include the excitonic effects, and only bare band structures were obtained by the LDA calculations. Even if both band structures were perfectly correct, the deviation could be caused by excitonic effects. Another possible explanation is the band gap correction that we employed in the comparison. We shifted the LDA band by ± 0.15 eV to correct for the underestimation of the bandgap, though this bandgap correction seems overly simple. In the future, the underestimation of the bandgap in LDA could be corrected by a heavier but more precise GW approximation. Precise calculations would provide a better comparison between the experimental band analysis and theoretical calculations. Furthermore, we must account for environment effects[43,44]. In our experiments, the SWCNTs were in water containing surfactants, while the SWCNTs in the DFT calculations were in a vacuum. Furthermore, the optical transition energies of SWCNT are highly modified by environmental effects, which were not considered in the DFT calculation or in our CD analysis. Although we must note the potential causes of these deviations, the band structures obtained by our CD analysis were far from those of the tight binding model and generally showed good agreement with the DFT bands. Though the pure experimental band analysis is the first challenge, the crossover of band energy observed in this study is helpful for understanding the DFT calculations because the calculated DOS does not explain the band index in the high energy region. We believe our simple analysis method may provide a tool for the experimental analysis of the unclear band structure of SWCNTs.

## Discussion

Because the band structure of a SWCNT is basically determined by a projection from the two dimensional graphene band to the one-dimensional SWCNT band, the band structure of a SWCNT has a systematic trend depending on its chiral structure. Usually, only the family behaviour of $E_{ii}$ transitions is discussed because the usual optical measurements can only give information regarding $E_{ii}$. However, the CD analysis allows for a discussion of the family behaviour of $E_{ij}$ transitions. The experimental $E_{ii}$ and $E_{ij}$ transition energies of the 12 (n,m) species obtained in this work are plotted as a function of the diameter ($d_t = 0.142\sqrt{3(n^2 + mn + m^2)}/\pi$; Supplementary Fig. 11). In agreement with the previous report[38], the transition energy crossover of $E_{ii}$ ($E_{33} > E_{44}$) was observed in some of Type 2 species such as (6,4), (9,1), (8,3), (10,2) and (9,4). Interestingly, further crossover of the $E_{31}$ and $E_{42}$ ($E_{31} > E_{42}$) was observed in a few Type 2 species, (9,1), (8,3) and (10,2) for the first time. These crossovers are known to be caused by a trigonal warping effect[9,10,39]. In the previous works, a simple scaling behaviour for the $E_{ii}$ optical transitions have been studied using an empirical formula shown in the following[45–47],

$$\mathrm{E}_{ii}(p_{ii}, d_t) - \beta_{pii}\cos3\theta/d_t^2 = a\frac{p_{ii}}{d_t}\left[1 + b\log\frac{c}{p_{ii}/d_t}\right] \quad (1)$$

where $\theta$ indicates a chiral angle, $p_{ii}$ indicates the cutting line index for $E_{ii}$ transition; $p_{ii} = 1$, 2, 4 and 5 for $E_{11}$, $E_{22}$, $E_{33}$ and $E_{44}$, respectively. To examine our assigned $E_{ii}$ and $E_{ij}$ transitions, the similar analysis was performed. First, we confirmed the same nonlinear scaling behaviour of $E_{ii}$ transitions obtained in this work (Supplementary Fig. 12a). Because resonance energies of Raman scattering was used for the fitting in the previous work, $E_{ii}$ values are limited in the range of excitation laser. The $E_{ii}$ values obtained in this work are in the different diameter range from them but the fitting parameters are almost same and all data are on the same curve. For the analysis of $E_{ij}$ transitions, we tried to extend Eq. 1 as follows.

$$\mathrm{E}_{ij}(p_{ij}, d_t) - \beta_{pij}\cos3\theta/d_t^2 = a\frac{p_{ij}}{d_t}\left[1 + b\log\frac{c}{p_{ij}/d_t}\right] \quad (2)$$

Here, the cutting line index $p_{ij}$ for $E_{ij}$ transition is 1.33, 1.67, 2, 3, 3 and 4 for $E_{12}$, $E_{21}$ $E_{13}$, $E_{31}$, $E_{24}$ and $E_{42}$, respectively. After careful optimization, we found that $E_{ij}$ transitions also exhibit a similar scaling behaviour with $E_{ii}$ transitions although the higher transitions have rather large deviations (Supplementary Fig. 12b). This means that the empirical formula is also available for $E_{ij}$ transitions.

Figure 6a–c shows the $E_{ii}$ and $E_{ij}$ transition energies estimated from the CD analysis as a function of $1/d_t$ (denoted as solid circles and solid triangles). The transition energies calculated from Eq. 1

and Eq. 2 are also shown for comparison (denoted as open circles and open triangles). The empirical calculation well reproduced all experimental values including their family behaviours. For example, $E_{21}$ is always larger than $E_{12}$ due to large electron-hole asymmetry. Interestingly, chiral angle dependence of $E_{12}$ is very small compared with the other transitions. Type dependences are also observed not only in $E_{ii}$ but also in $E_{ij}$. Even for higher transitions, we can expect the family behaviour. However, rather large deviations were observed especially for small diameter species. The largest error was 0.36 eV for $E_{24}$ transition of (7,3), which is probably due to the modified band structure by a large curvature effect. Because of the good agreement between the experimental and calculated values, $E_{ij}$ values can be estimated for other species that are not measured. Although the exciton effects are not well known especially for higher lying transitions, our findings firstly reveal that the diameter, chiral angle, type play an important role in the electron-hole asymmetry of SWCNTs, which is very useful for deeper understanding of the physical properties of SWCNT.

In conclusion, we developed an advanced method for separating single-chirality SWCNT enantiomers by column chromatography in combination with overloading selective adsorption and stepwise elution using a mixed surfactant. This separation afforded the highest purity of both single-chirality samples and enantiomeric SWCNTs obtained thus far. The high-purity single-chirality SWCNTs enantiomers enabled us to analyse their complicated CD spectra. We could assign $E_{ii}$ and $E_{ij}$ optical transitions to the CD peaks by considering the selection rules of optical transitions, the alternating CD signs for $E_{ii}$ transitions, the crossovers of $E_{ii}$ transitions, the relationships between $E_{ii}$ and $E_{ij}$ (for example, $(E_{11} + E_{22})/2 = (E_{12} + E_{21})/2$), and other factors. Using the obtained $E_{ii}$ and $E_{ij}$ energies, the asymmetric excitonic band structures of the 12 different (n,m) SWCNTs were successfully determined. This is the first experimental determination of asymmetric band structures of an extensive set of (n,m) SWCNTs. Furthermore, based on the present experimental results and empirical formula, we demonstrated $E_{ij}$ for many (n,m) species and revealed (n,m) dependent electron-hole asymmetry.

## Methods

**Preparation of SWCNT dispersion.** SWCNTs produced by high-pressure catalytic CO decomposition (HiPco, Raw, 0.8 ∼ 1.2 nm in diameter, NanoIntegris, Skokie, Illinois, USA) were used as a starting material. A net weight of 100 mg of SWCNT powder was dispersed in 100 ml of 3% SDS (ultra-pure ≥99.0%, Sigma-Aldrich) aqueous solution in a glass bottle by an ultrasonic homogenizer (output 30%, Sonifier 250D, Branson) for 20 h (all surfactant concentrations were weight percentage). The bottle was immersed in a water bath at 20 °C. After ultrasonication, the solution was ultracentrifuged for 2 h to sediment bundles and impurities (S50A rotor, $210,000 \times g$, 25 °C, CS150GX, Hitachi Koki, Tokyo, Japan). The upper 80% of the supernatant was collected as the isolated SWCNT dispersion and was used for the following separation.

**Chirality and enantiomer separation.** The first separation utilized the overloading selective adsorption method. Five open columns filled with 5.6 ml of gel (Sephacryl S-200, GE Healthcare) were vertically connected. After equilibration with 3% SDS solution, 80 ml of the SWCNT dispersion was loaded into the top of the multicolumn. Because the amount of SWCNTs was much larger than the capacity of the gel, excess SWCNTs flowed through the bottom column (the unadsorbed fraction). The excess SWCNTs were recovered for the next round of overloading selective adsorption. After a wash with 3% SDS solution, the multicolumn was separated into its individual columns. The adsorbed SWCNTs were eluted with 1% SC (ultra-pure ≥99.0%, Sigma-Aldrich) solution, producing fractions from Col.1 to Col.5. Six rounds of the overloading process using the unadsorbed fraction were repeated, and fractions from Col.1 to Col.30 were obtained (Supplementary Fig. 1). Then, the second separation utilizing a stepwise elution with a mixed surfactant was conducted. The same amount of 1% SDS solution was added to the fractions (Col.1, 6, 11, 16 or 26). The resulting SWCNTs in 0.5% SC + 0.5% SDS solution were injected into a column filled with 5.6 ml of the gel equilibrated with 0.5% SC + 0.5% SDS solution, followed by washing with the same solution. The SWCNTs adsorbed onto the gel were eluted

by a stepwise increase in the concentration of DOC (purity ≥96.0%, Wako Pure Chemical Industries, Osaka, Japan) in the 0.5% SC + 0.5% SDS solution; the concentration of DOC was successively increased from 0.01 to 0.07% with 0.01 or 0.005% steps (Supplementary Fig. 2). Separation processes were performed at room temperature (23 ± 0.5 °C). The detailed methods and results of the separation under different SDS concentrations in the first separation and of the separation of larger diameter enantiomers are given in the Supplementary Methods, Figs 3 and 4.

**Optical measurements.** Optical absorption spectra were measured from 1,400 to 200 nm in 2 nm increments using an ultraviolet–visible near-infrared spectro-photometer (UV-3600, Shimadzu, Kyoto, Japan). Photoluminescence spectra were measured using a spectrofluorometer (Nanolog, HORIBA, Kyoto, Japan) equipped with a liquid nitrogen-cooled InGaAs near-infrared array detector. The excitation wavelength was varied from 400 to 800 nm in 5 nm steps, and the emission wavelength was varied from 791 to 1304 nm in 1 nm increments. The spectral slit widths were 7 nm for both the excitations and emissions. All raw intensities were corrected with the intensities of the lamp spectrum. The CD spectra of the sorted single-chirality enantiomers were measured from 900 to 200 nm in 1 nm increments using a CD spectropolarimeter (J-820, JASCO, Tokyo, Japan).

**Date availability.** The data that support the findings of this study are available from the corresponding author on request.

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

## Acknowledgements

We thank Stephen K. Doorn (Los Alamos National Laboratory), Shigeo Maruyama (The University of Tokyo) and Yuhei Miyauchi (Kyoto University) for helpful discussions on the band structure analysis of SWCNTs. R.S. acknowledges MEXT Grant (Nos. 25286005, 25107005). This work was supported by JSPS KAKENHI, grant no. 25220602.

## Author contributions

X.W. performed all the experiments and data analysis. X.W., T.T., Y.Y. and H.K. contributed to the separation and the discussion of the results. N.S. and R.S. contributed to the theoretical calculation of the CD spectra. X.W., T.T. and H.K. contributed to the discussion of the band structure and family behaviour. X.W. wrote the paper. T.T. and H.K. reviewed and corrected the paper.

## Additional Information

**Competing financial interests:** The authors declare no competing financial interests.

