## [Peer review file · Nature Communications]

Reviewers' Comments:

Reviewer #1 (Remarks to the Author)

The authors report experimental circular dichroism (CD) spectra on 12 different high-purity single-chiral single-walled carbon nanotube enantiomers. The measured spectra was so perfect, that they could extract the excitonic band structures in most of the cases, for the first time. They also compare the obtained results to density functional theory calculation.

In the experimental analysis they used gel column chromatography that combined overloading selective adsorption with stepwise elution to separate 12 different high-purity single-chiral SWCNT enantiomers. This is the continuation of the work done in Ref 14 of the manuscript with overlapping authors (TT and HK). After fine tuning of the process, the samples showed much higher CD intensities than the highest values reported in the previous works, indicating the highest enantiomeric purity of the samples.

The experimental results are of interest, the described method is nice and efficient. The experimental methods and data is well-presented.

However, I have reservations about the theory presented in the manuscript.

- "From the definition of the P- and M-enantiomers, the E11~E55 signals of P-enantiomers exhibit a (+,-,+,-,+) progression, whereas the signals of the M-enantiomers exhibit the inverted (-,+,-,+,-) progression⁹."

I find this statement non-trivial and could not find an explanation for the alternating behaviour. In Ref 9 I could not find this, and I tried and failed to find it in the referenced theoretical paper (Ref 11), too. I also miss an explanation how is it related to the fact, that it is a Type I or II SWCNT. Instead of the rather trivial figure 4b, I would welcome a figure of the BZ and cutting lines in Type I and II cases indicating the correct (alternating) sign of CD.

This paper (<http://pubs.acs.org/doi/abs/10.1021/ja062095w>) contains a similar statement based on Ref 5 (<http://journals.aps.org/prb/abstract/10.1103/PhysRevB.57.9301>).

- I could not find the details of the presented LDA calculations (cutoff energy, K-set, etc.). The corresponding Ref 6 does not contain data for most for these SWCNTs. I suggest to clarify this either by adding the details to the Supp. Mat. or the correct reference.

- "In the future, the underestimation of the band gap in LDA could be corrected by a heavier but more precise GW approximation." -- There are GW results available for some SWCNTs for 10 years, e.g.:

Phys. Rev. Lett. 92, 077402 (2004). I wonder, if based on the available GW results a more sophisticated scaling of the LDA calculation is possible to this case.

Reviewer #2 (Remarks to the Author)

Xiaojun et al. used circular dichroism to measure the optical transitions of diameter-, chiral angle- and enantiomer-sorted single wall carbon nanotubes (SWCNTs). The scientific novelty is the assignment of many E_{ii} and E_{ij} transitions from several specific SWCNTs, allowing the determination of the excitonic band structure of carbon nanotubes, including the asymmetry between valence and conduction bands, which is missing in the related literature. The work, as idealized, is very interesting and important. However, in my opinion the authors do not deliver what is promised. The highest weakness is the formalism to conclude about the (n,m)-dependent excitonic band structure. The authors compare their results with DFT calculations, which is known to be unappropriated. Since the authors are not working to deliver information about the accuracy of state-of-the-art calculations involving DFT+GW+BSP, they should at least provide empirical formula, such as in [Araujo et al. Physica E 42, 1251-1261 (2010)], and discuss how diameter and

chiral angle play a role in the electron-hole asymmetry. Since at the end the authors do not report any pair ($E_{ij}+E_{ji}$), I wonder if this is achievable with their results. Without this information and discussion, I don't think this paper has enough new science to deserve publication, since the effects themselves have already been published, as properly cited by the authors.

To help focusing their work, I have some other criticisms the authors could consider:

- The description of sample preparation is too long, including 2 tables and 3 figures. The authors could move most of this material to the supplemental information. Maybe Table 2 could be translated into the size of the triangles in Fig.4a.
- For completeness, the labelling "left-handed" is missing in Fig.4b.
- Figure 6 does not deliver too much information considering Fig.2, and the information contained in Fig.S8 is considerably important to be out. The authors should reevaluate this data and data-analysis presentation.
- Fig.7 is interesting, but again, at that point of the development of SWCNT photophysics, it makes no sense to compare their optical transitions with DFT calculations. This figure could be in the supplementary information as a curiosity.
- Fig.8 should be better explored to generate an empirical formula for E_{ii} and E_{ij} that should show consistency with previous works, such as the review [Araujo et al. *Physica E* 42, 1251-1261 (2010)].

Reviewer #3 (Remarks to the Author)

This is definitely nice work describing the separation and CD spectroscopy of SWCNTs. The authors used the CD spectra together with selection rules to determine excitonic band structures of SWCNTs. I am not absolutely convinced that the work is suitable for *Nature Communications*. It would probably be better suited for a more specialized journal. In any case there are some points that need to be addressed before publication, as outlined below.

I believe that the paper reports the most complete and highest purity of chiral enantio-separated SWCNTs samples. However, the methods were reported before so the preparation of the samples is "just" an extension of previous work.

Specific comments:

-There is no doubt that the samples are very pure, most of them are of higher purity than previously reported. However there is a weak point here that needs to be addressed / discussed. The authors quantify the purity of (n,m) species from the integrated intensity ratio of the absorption peaks. The analysis described assumes that the absorption cross sections are identical for all (n,m) species, which is not the case (*Nano Lett.* 2014, 14, 1530–1536).

-Similarly, the authors use a normalized CD signal (CD_{norm}) to assess the enantiomeric purity of the samples. This is fine, but why do the authors use such a strange measure. Why not use the anisotropy factor $\Delta A/A$ (which is related to CD_{norm}), which is commonly used in CD spectroscopy?

-The authors use P and M to refer to the enantiomers that showed plus or minus E11 CD signals. This is very misleading since P and M are reserved to describe the absolute configuration of molecules with helical chirality. The authors need to use other names!

-The authors speculate about the absolute configuration (handedness) of the SWCNTs on page 11 based on affinity arguments. This is very dangerous and highly speculative. The best thing to do would be to calculate the CD spectrum in order to assign the handedness. At least the authors should modify this part of the text.

Reply to Reviewers' comments for

“Experimental Determination of Excitonic Band Structures of Single-Wall Carbon Nanotubes using Circular Dichroism Spectra”

by Xiaojun Wei, Takeshi Tanaka, Yohei Yomogida, Naomichi Sato, Riichiro Saito, and Hiromichi Kataura

Response to Reviewer #1:

The authors report experimental circular dichroism (CD) spectra on 12 different high-purity single-chiral single-walled carbon nanotube enantiomers. The measured spectra was so perfect, that they could extract the excitonic band structures in most of the cases, for the first time. They also compare the obtained results to density functional theory calculation.

In the experimental analysis they used gel column chromatography that combined overloading selective adsorption with stepwise elution to separate 12 different high-purity single-chiral SWCNT enantiomers. This is the continuation of the work done in Ref 14 of the manuscript with overlapping authors (TT and HK). After fine tuning of the process, the samples showed much higher CD intensities than the highest values reported in the previous works, indicating the highest enantiomeric purity of the samples.

The experimental results are of interest, the described method is nice and efficient. The experimental methods and data is well-presented.

Reply:

We thank you for reviewing our manuscript and for giving careful assessment on our manuscript. We are very glad for your interest in our experimental method and results.

Comment 1:

However, I have reservations about the theory presented in the manuscript.

- "From the definition of the P- and M-enantiomers, the E11~E55 signals of P-enantiomers exhibit a (+,-,+,-,+) progression, whereas the signals of the M-enantiomers exhibit the inverted (-,+,-,+,-) progression⁹."

I find this statement non-trivial and could not find an explanation for the alternating behaviour. In Ref 9 I could not find this, and I tried and failed to find it in the referenced theoretical paper (Ref 11), too.

I also miss an explanation how is it related to the fact, that it is a Type I or II SWCNT. Instead of the rather trivial figure 4b, I would welcome a figure of the BZ and cutting lines in Type I and II cases indicating the correct (alternating) sign of CD.

This paper (<http://pubs.acs.org/doi/abs/10.1021/ja062095w>) contains a similar statement based on Ref 5 (<http://journals.aps.org/prb/abstract/10.1103/PhysRevB.57.9301>).

Reply 1:

Thank you for your valuable comment. Experimentally, alternating behavior of E_{ii} CD signal has been clearly observed in this work and also in some previous reports. Our previous manuscript was based on this empirical rule. However, as the reviewer pointed out, there was no clear theoretical evidence of this alternating behavior. We agreed that the theoretical background is necessary before the detailed analysis of CD spectra. To clarify

this point, we added two authors and calculated CD spectra of SWCNTs theoretically. We found that the CD signal of chiral SWCNTs which corresponds to the difference of optical absorption between left and right circularly polarized light, is originated from the phase difference of the incident circularly polarized light for different carbon atoms in SWCNT. Since the method for calculating the CD spectra is rather complicated for this experimental paper, we only show the calculated results in this paper for simplicity. As a result, the alternating behavior of E_{ij} CD signals which is proportional to d_i/λ (diameter/wavelength) was clearly demonstrated. Furthermore, the opposite CD sign between Type 1 and Type 2 CNT was also demonstrated. These systematic alternating behaviors come from the sign of wavefunctions on one-dimensional cutting line near the K point. These theoretical results are consistent with the experimental observations at E_{ij} energies even though we did not consider the exciton effect. Since the exciton wavefunction is expressed by linear combination of the Bloch wavefunctions near the k_{ij} point, the alternating behavior of the calculated CD spectra will not change if we consider the exciton wavefunction. To emphasize this point, as the suggestion of the reviewer, we have removed the previous Figure 4b and added calculated CD spectra of enantiomers for Type 1 and Type 2 SWCNTs, a figure of the BZ and cutting lines, and the related discussion in the revised manuscript.

Figure 4. Theoretical calculation of CD spectra for enantiomers.

(a,b) Calculated CD signal ΔW of enantiomers for (a) (6,5) (red) and (11,-5) (blue) and (b) (7,5) (red) and (12,-5) (blue), as a function of wavelength of incident light. Shaded lines correspond to E_{ii} and E_{ij} van Hove singularity of joint density of states. (c) Reciprocal space of the graphene and cutting lines corresponding to Type 1 (left) and Type 2 (right) SWCNTs.

Previous:

However, our results showed that the separation order of M- and P-enantiomers depended on the type of SWCNT index. Our measurement could not determine which enantiomer was right-handed; thus, we might assume that right-handed SWCNTs had lower affinities for the gel and eluted at lower DOC concentrations. Based on this assumption, P-enantiomers could be identified as right-handed SWCNTs for Type 1 SWCNTs and left-handed SWCNTs for Type 2 SWCNTs, and vice versa. More simply, the CD signals of Type 1 and Type 2 SWCNTs have opposite signs. This phenomenon can be understood as a difference in the cutting lines for Type 1 and Type 2 around the K point in the Brillouin zone.

Revised:

Page 11, line 192:

However, our results showed that the separation order of (-)- and (+)-enantiomers depended on the type of SWCNT index. A possible explanation is that the CD signals of Type 1 and Type 2 SWCNTs with a specific handedness have opposite signs. This statement is supported by theoretical calculation of CD spectra of SWCNTs described below.

Previous:

Figure 4b illustrates the CD signal at E_{22} of left-handed and right-handed SWCNTs for Type 1 and Type 2 SWCNTs. Our finding suggests that the previous definition, namely, that SWCNTs exhibiting a positive CD peak at E_{22} can be identified as left-handed enantiomers and denoted as M, is not always correct^{9,13,14}; instead, the identification of left-handed enantiomers depends on the type of the SWCNT. The finding is supported by the theoretical study of CD spectra of SWCNTs by first-principles calculation¹¹.

Revised:

Page 12, line 207:

In Figure 4a, we show the calculated CD signal ΔW as a function of wavelength of the incident circular polarized lights in the direction perpendicular to the nanotube axis, for (6,5) (red line) and (11,-5) (blue line) SWCNTs in which (n,m) and $(n+m,-m)$ SWCNTs are enantiomers to each other. In the numerical calculation, ΔW is defined by the difference of optical absorption probability of (6,5) (or (11,-5)) SWCNT for the left- and right-handed circular polarized lights within tight-binding method for optical transition of SWCNTs³². It is important to note that the optical absorption probability near the K point for the right-handed circular polarized light is identical to that near the K' point for the left-handed circular polarized light because of the time reversal symmetry between the K and K' points, if we adopt an approximation of the vertical optical transition in the k space. It means that we would get $\Delta W = 0$ if we adopted this approximation. Thus it is essential in the calculation of ΔW to consider the *phase* difference of the light for different carbon atom positions that is usually neglected in the calculation of the absolute values of the optical absorption. The detail of the calculation will be reported elsewhere.

The calculated results reproduce the experimental results in many aspects: (1) ΔW of (6,5) SWCNT changes the sign at E_{22} , E_{33} , and E_{44} , and (2) ΔW of (11,-5) SWCNTs gives the opposite sign to that of (6,5)

for all the wavelength of the light. Further, when we change (n,m) from (6,5) (Type 1 SWCNT) to (7,5) (Type 2 SWCNT) as shown in Figure 4b, we get the opposite sign of ΔW to that for (6,5) for each E_{ii} energy. This phenomenon can be explained by the difference in the reciprocal space of the graphene and cutting lines between Type 1 and Type 2 SWCNTs as shown in Figure 4c. The optical absorption for E_{ii} van Hove singularity of Type 1 SWCNT occurs at the opposite k points measured from the K point compared to that of Type 2 SWCNT, whose transition optical dipole moment around the K point gives the opposite sign³³. The opposite behavior between type 1 and 2 SWCNTs in the calculation is also consistent with the experimental expectation based on their type-dependent affinities with gel. Thus alternative behavior of ΔW as a function of the wavelength comes from the change of the wavefunctions on one-dimensional cutting line near the K point^{34,35}.

It should be mentioned that we measure CD of SWCNTs with random orientation in the experiment. For incident circular polarized light, there are two inequivalent geometries of SWCNTs: The propagating directions of the light are (1) perpendicular to the nanotube axis (the case of Figure 4a and 4b) and (2) parallel to the nanotube axis (not shown in this paper). Even though we should integrate the CD for possible direction of the light, we did not show this for simplicity. We can imagine that the case of (2) gives a minor contribution after integration of azimuth angle of spherical coordinate. Further, we did not consider the effect of the exciton in this calculation in the first approximation. Since the wavefunction of an exciton is given by linear combination of the Bloch functions along one-dimensional Brillouin zone (or cutting line)^{36,37}, we can say that the behaviors of CD do not change even after we consider the exciton effect. Thus the present calculated results are reasonable to simulate the experimental results.

Previous:

Furthermore, the CD signal for E_{ii} transitions should alternate by either increasing or decreasing by one order. For example, if the CD of E_{11} is positive, the CD of E_{22} is negative. From the definition of the P- and M-enantiomers, the $E_{11}\sim E_{55}$ signals of P-enantiomers exhibit a (+,-,+,-,+) progression, whereas the signals of the M-enantiomers exhibit the inverted (-,+,-,+,-) progression⁹.

Revised:

Page 15, line 264:

Furthermore, **based on this alternating behavior of E_{ii} CD signals** of the (+)- and (-)-enantiomers, the $E_{11}\sim E_{55}$ signals of (+)-enantiomers exhibit a (+,-,+,-,+) progression, whereas the signals of the (-)-enantiomers exhibit the inverted (-,+,-,+,-) progression.

Comment 2:

- I could not find the details of the presented LDA calculations (cutoff energy, K-set, etc.). The corresponding Ref 6 does not contain data for most for these SWCNTs. I suggest to clarify this either by adding the details to the Supp. Mat. or the correct reference.

Reply 2:

Thank you for your careful reading. We are very sorry for that we missed a necessary reference: group web page of Prof. Susumu Saito (corresponding author of Ref. 6), http://www.stat.phys.titech.ac.jp/saito/optCNTs/OptCNT_LDA_DOS_Kato.html, where the details of the DOS for (n,m) species presented in our manuscript are shown. We added this web page as a reference in the

revised manuscript.

Comment 3:

- "In the future, the underestimation of the band gap in LDA could be corrected by a heavier but more precise GW approximation." -- There are GW results available for some SWCNTs for 10 years, e.g.: *Phys. Rev. Lett.* 92, 077402 (2004). I wonder, if based on the available GW results a more sophisticated scaling of the LDA calculation is possible to this case.

Reply 3:

Thank you for your important comment. As you pointed out, Susumu Saito has already compared GW result with LDA one for (7,0) and some other SWCNTs (*Phys. Rev. B* 68, 155424 (2003)). Their conclusion was that the considerable enlargement of bandgap was necessary to fit the both calculations. To discuss large diameter SWCNT, they also calculated band structure of graphene as an infinite diameter SWCNT. The result showed that 13% expansion of band dispersion is enough to get consistency. In our case, diameters of separated SWCNTs are not far from (7,0). According to Saito's results, therefore, we changed bandgap correction for LDA calculation from 20% expansion of band dispersion to 0.3 eV bandgap enlargement. We have revised the data on the comparison (Supplementary Figure 10). The DOS spectra are slightly different from the previous ones but the agreement or disagreement is not changed so much. We compared the bandgap corrected LDA results and our experimentally obtained energy levels again. Since we modified LDA bandgap to fit the excitonic band gap, it is trivial that 1_v and 1_c levels show good agreement. When we look at 2_v and 2_c levels, we can see good agreements for (6,4), (8,3), and (10,2). In most of species, interestingly, we found redshift of both 2_v and 2_c levels keeping E_{22} value. Since we used E_{22} and E_{21} CD peak energies to get 2_v and 2_c levels, the redshift in 2_v and 2_c levels from LDA band is probably caused by the difference in exciton binding energy between E_{21} and E_{22} transitions. To emphasize this point, we have add following arguments in the main manuscript.

Supplementary Figure 10. Comparison of experimental results based on CD spectra (red dash line) and DOS calculated by LDA^{3,4} (blue solid line). The energy bands (n_c and n_v) in the LDA calculation were shifted away from the Fermi level by +0.15 and -0.15 eV for comparing, respectively.

Previous:

For comparison, the electronic density of states (DOS) was calculated by DFT⁶ using local density approximation (LDA); these results are indicated by solid blue curves. Here, the binding energy values of the DFT results are expanded by 20% to correct for the underestimation of the bandgap energy. Upon first observation, all SWCNTs appear to have asymmetric valence and conduction bands, which is roughly consistent with the DFT calculation. The highly chiral angle species (6,4), (6,5), (7,3), (7,5), (9,4) and (7,6) showed strong agreement with the DFT calculations. However, the low chiral angle species, such as (9,1), (8,3), (10,2), (8,4), (10,3) and (11,1), showed rather large deviations, especially in the conduction bands. These deviations were probably caused primarily by exciton effects.

In the future, the underestimation of the bandgap in LDA could be corrected by a heavier but more precise GW approximation.

Revised:

Page 19, line 324:

The experimental results based on CD spectra were compared with the electronic density of states (DOS) calculated by DFT using local density approximation (LDA)^{6,7} (see Supplementary Figure 10). Generally, GW approximation gives more precise band structure including bandgap⁴¹. However, GW calculations were limited for very small diameter SWCNTs to date due to required huge resources. On the other hand, Saito et al. have already published DFT band structures for most of SWCNTs using LDA^{6,7}. They also compared LDA and GW calculations for (7,0) SWCNT and found that well known bandgap underestimation in LDA can be corrected by simple bandgap enlargement⁴². Therefore, here we corrected DFT results by shifting n_c and n_v away from the Fermi level for +0.15 and -0.15 eV, respectively. Upon first observation, all SWCNTs appear to have asymmetric valence and conduction bands, which is roughly consistent with the DFT calculation. Since we modified LDA bandgap to fit the excitonic band gap, it is trivial that 1_v and 1_c levels show good agreement for most of species. Especially, the species (6,4), (8,3), and (10,2) showed good agreement in the second conduction and valence bands (2_c and 2_v). However, (6,5), (7,5), (9,4), (8,4), (7,6), (10,3), and (11,1) showed redshift of both 2_c and 2_v bands keeping good agreement for E_{22} . Since E_{22} and E_{21} CD peaks were used to estimate asymmetric 2_v and 2_c levels, the redshift in 2_v and 2_c levels from LDA band is probably caused by the difference in exciton binding energy between E_{21} and E_{22} transitions. On the other hand, the species (9,1), and (7,3) disagreed with E_{22} value from DFT calculation. This suggests that the exciton binding energies of E_{11} and E_{22} are different in these species. Most of species showed rather large deviations in the higher order energy bands ($n_c, n_v \geq 3$) probably due to more complicated excitonic and environmental effects.

Previous:

We expanded the LDA band by 20% to correct for the underestimation of the bandgap, though this bandgap correction seems overly simple.

Revised:

Page 20, line 350:

We shifted the LDA band by ± 0.15 eV to correct for the underestimation of the bandgap, though this bandgap correction seems overly simple.

Response to Reviewer #2:

Xiaojun et al. used circular dichroism to measure the optical transitions of diameter-, chiral angle- and enantiomer-sorted single wall carbon nanotubes (SWCNTs). The scientific novelty is the assignment of many E_{ii} and E_{ij} transitions from several specific SWCNTs, allowing the determination of the excitonic band structure of carbon nanotubes, including the asymmetry between valence and conduction bands, which is missing in the related literature. The work, as idealized, is very interesting and important.

Reply:

We thank you for reviewing our manuscript and giving careful assessment and constructive recommendations on our manuscript.

Comment 1:

However, in my opinion the authors do not deliver what is promised. The highest weakness is the formalism to

conclude about the (n,m)-dependent excitonic band structure. The authors compare their results with DFT calculations, which is known to be unappropriated. Since the authors are not working to deliver information about the accuracy of state-of-the-art calculations involving DFT+GW+BSP, they should at least provide empirical formula, such as in [Araujo et al. Physica E 42, 1251-1261 (2010)], and discuss how diameter and chiral angle play a role in the electron-hole asymmetry.

Reply 1:

Thank you very much for your constructive comments. We agree with your opinion. The analysis based on the empirical formula is very useful for taking an overview of the electron-hole asymmetry. We tried further analysis of the both scaling behaviors of E_{ii} and E_{ij} transitions based on the empirical formula. As a result, the observed scaling behaviors of E_{ii} transitions agree with those in previous reports. Interestingly, E_{ij} transitions also exhibit a similar scaling behavior with E_{ii} transitions. Using the empirical formulas for E_{ij} transitions, we have demonstrated E_{ij} for many (n,m) species (see Figure 6b and 6c in the revised manuscript). Finally, (n,m) dependent electron-hole asymmetry was discussed.

Additionally, through this analysis, an incorrect assignment was found in E_{55} transition of (6,5) SWCNT. We have corrected it from 5.30 to 4.32 eV (see Figure for Reviewer shown in the following). The assignments of E_{42} of (10,2), and E_{21} and E_{31} of (11,1) were also revised. Related results are shown in Supplementary Figure 9 and 11b.

Corresponding revision will be shown after comment 2.

Figure for Reviewer. Nonlinear scaling behavior of E_{ii} transitions.

Comment 2:

Since at the end the authors do not report any pair ($E_{ij}+E_{ji}$), I wonder if this is achievable with their results. Without this information and discussion, I don't think this paper has enough new science to deserve publication, since the effects themselves have already been published, as properly cited by the authors.

Reply 2:

The pair of E_{ij} and E_{ji} (described as E_{ij} in the revised manuscript) is achievable from our results. They are used for further analysis of the family behavior of obtained E_{ij} transitions by combining the empirical formula (as reply for Comment 1).

Previous: (none)

Revised: (for Comment 1 and 2)

Page 2, line 22:

Finally, we demonstrated that an extended empirical formula can estimate E_{ij} optical transition energies for any (n,m) species.

Page 24, line 429:

Furthermore, based on the present experimental results and empirical formula, we demonstrated E_{ij} for many (n,m) species and revealed (n,m) dependent electron-hole asymmetry.

Previous:

If our assignments were correct, we should have observed a smooth family pattern in the E_{ij} transitions such as that seen for the E_{ii} . We believed this plot could be used as a passive check of the accuracy of our CD analysis. Figure 8a shows the excitation energy plots for the E_{ii} (E_{11} – E_{44}) transitions of the 12 obtained (n,m) species. In agreement with the theoretical calculations^{29,33,34}, the transition energy crossover, $E_{33} > E_{44}$, was observed in the Type 2 SWCNTs (6,4), (9,1), (8,3), (10,2), and (9,4). Figure 8b shows the excitation energy of the E_{ij} (E_{21} , E_{31} and E_{42}) transitions. The dashed curves indicate a family pattern of E_{ij} energies that is visually appealing, and the systematic changes of the E_{ij} values in each transition suggested that our assignments were reasonably correct. Similar to the crossover of the E_{33} and E_{44} transitions, further crossovers of the E_{31} and E_{42} transitions were observed for (9,1), (8,3), and (10,2). These results were probably due to a trigonal warping effect^{9,10,29}. Interestingly, the E_{ij} transitions had very similar family characteristics as the E_{ii} transitions, suggesting that the E_{ij} value mainly depended on the depth of the valence band. Because the conduction band width was narrower than that of the valence band, the difference in conduction band energy did not substantially affect the family behaviors. This result is reasonable, and these results will provide an important reference for further studies of cross-polarized optical measurements.

Revised:

Page 21, line 372:

The experimental E_{ii} and E_{ij} transition energies of the 12 (n,m) species obtained in this work are plotted as a function of the diameter ($d_t = 0.142\sqrt{3(n^2 + mn + m^2)} / \pi$) (see Supplementary Figure 11). In agreement with the previous report³⁸, the transition energy crossover of E_{ii} ($E_{33} > E_{44}$) was observed in some of Type 2 species such as (6,4), (9,1), (8,3), (10,2), and (9,4). Interestingly, further crossover of the E_{31} and E_{42} ($E_{31} > E_{42}$) was observed in a few Type 2 species (9,1), (8,3), and (10,2) for the first time. These crossovers are known to be caused by a trigonal warping effect^{10,11,39}. In the previous works, a simple scaling behavior for the E_{ii} optical transitions have been studied using an empirical formula shown in the following⁴⁵⁻⁴⁷,

$$E_{ii}(p_{ii}, d_t) - \beta_{p_{ii}} \cos 3\theta / d_t^2 = a \frac{p_{ii}}{d_t} \left[1 + b \log \frac{c}{p_{ii} / d_t} \right] \quad (1)$$

where θ indicates a chiral angle, p_{ii} indicates the cutting line index for E_{ii} transition; $p_{ii} = 1, 2, 4, 5$ for $E_{11}, E_{22}, E_{33}, E_{44}$, respectively. To examine our assigned E_{ii} and E_{ij} transitions, the similar analysis was performed. Firstly, we confirmed the same nonlinear scaling behavior of E_{ii} transitions obtained in this work (see

Supplementary Figure 12a). Because resonance energies of Raman scattering was used for the fitting in the previous work, E_{ii} values are limited in the range of excitation laser. The E_{ii} values obtained in this work are in the different diameter range from them but the fitting parameters are almost same and all data are on the same curve. For the analysis of E_{ij} transitions, we tried to extend Eq. 1 as follows.

$$E_{ij}(p_{ij}, d_t) - \beta_{p_{ij}} \cos 3\theta / d_t^2 = a \frac{p_{ij}}{d_t} \left[1 + b \log \frac{c}{p_{ij}/d_t} \right] \quad (2)$$

Here, the cutting line index p_{ij} for E_{ij} transition is 1.33, 1.67, 2, 3, 3, and 4 for E_{12} , E_{21} , E_{13} , E_{31} , E_{24} , and E_{42} , respectively. After careful optimization, we found that E_{ij} transitions also exhibit a similar scaling behavior with E_{ii} transitions although the higher transitions have rather large deviations (see Supplementary Figure 12b). This means that the empirical formula is also available for E_{ij} transitions.

Figure 6a-c show the E_{ii} and E_{ij} transition energies estimated from the CD analysis as a function of $1/d_t$ (denoted as solid circles and solid triangles). The transition energies calculated from Eq. 1 and Eq. 2 are also shown for comparison (denoted as open circles and open triangles). The empirical calculation well reproduced all experimental values including their family behaviors. For example, E_{21} is always larger than E_{12} due to large electron-hole asymmetry. Interestingly, chiral angle dependence of E_{12} is very small compared with the other transitions. Type dependences are also observed not only in E_{ii} but also in E_{ij} . Even for higher transitions, we can expect the family behavior. However, rather large deviations were observed especially for small diameter species. The largest error was 0.36 eV for E_{24} transition of (7,3), which is probably due to the modified band structure by a large curvature effect. Because of the good agreement between the experimental and calculated values, E_{ij} values can be estimated for other species that are not measured. Although the exciton effects are not well known especially for higher lying transitions, our findings firstly reveal that the diameter, chiral angle, type play an important role in the electron-hole asymmetry of SWCNTs, which is very useful for deeper understanding of the physical properties of SWCNT.

Supplementary Figure 12. Nonlinear scaling behavior of E_{ii} and E_{ij} transitions.

(a) Experimental E_{ii} transition energies after correcting for the chirality dependence ($E_{ii}^{\text{exp}} - \beta_{p_{ii}} \cos 3\theta / d_t^2$), as a

function of p_{ij}/d_t . Gray symbols indicate the data from Suppl. Ref. 5 for comparison. Dotted curves indicate E_{ij} energies calculated by the empirical formula, $E_{ij}(p_{ij}, d_t) - \beta_{p_{ij}} \cos 3\theta / d_t^2 = a \frac{p_{ij}}{d_t} \left[1 + b \log \frac{c}{p_{ij}/d_t} \right]$, given in Suppl. Ref. 5. Parameters a , b , c , and $\beta_{p_{ij}}$ were determined from our experimental results by fitting. Red curve shows result for E_{11} and E_{22} where $a = 0.970$ eV·nm, $b = 0.497$, $c = 1.154$ nm⁻¹. Blue curve shows result for E_{33} and E_{44} transitions, where $a = 1.079$ eV·nm, $b = 0.668$, and $c = 1.467$ nm⁻¹. Inset shows the $\beta_{p_{ij}}$ values for the lower (upper) branch in the family pattern of $E_{ij} - d_t$ plot, -0.07 (0.05), -0.19 (0.17), -0.28 (0.42), -0.40 (0.42) for $p_{ij} = 1, 2, 4, 5$, respectively. (b) Experimental E_{ij} transition energies after correcting for the chirality dependence ($E_{ij}^{\text{exp}} - \beta_{p_{ij}} \cos 3\theta / d_t^2$), as a function of p_{ij}/d_t , where the p_{ij} values are 1.33, 1.66, 2, 3, 3, and 4 for E_{12} , E_{21} , E_{13} , E_{31} , E_{24} , E_{42} , respectively, from the cutting lines of these transitions. The E_{ij} transitions are fitted by the extended empirical formula, $E_{ij}(p_{ij}, d_t) - \beta_{p_{ij}} \cos 3\theta / d_t^2 = a \frac{p_{ij}}{d_t} \left[1 + b \log \frac{c}{p_{ij}/d_t} \right]$. Where, fitted parameters are $a = 0.932$ eV·nm, $b = 0.547$, and $c = 1.412$ nm⁻¹. Dotted curves indicate the fitting results for all E_{ij} transitions. Inset: the applied $\beta_{p_{ij}}$ values denoted as red circles for the lower (upper) E_{ij} branches are -0.04 (0.04), -0.10 (0.11), -0.20 (0.14), -0.22 (0.28), -0.30 (0.20), -0.32 (0.40) for E_{12} , E_{21} , E_{13} , E_{31} , E_{24} , E_{42} , respectively. The black circles indicate $\beta_{p_{ii}}$ for comparison.

Figure 6. E_{ii} and E_{ij} transition energies.

(a-c) Transition energy of E_{ii} and E_{ij} as a function of $1/d_i$. Open and solid symbols (circles for Type 1 and triangles for Type 2 species) indicate the calculation and experimental results, respectively.

Comment 3-1:

To help focusing their work, I have some other criticisms the authors could consider:

- The description of sample preparation is too long, including 2 tables and 3 figures. The authors could move most of this material to the supplemental information. Maybe Table 2 could be translated into the size of the triangles in Fig.4a.

Reply 3-1:

Thank you for your constructive comments. We have moved one Table and one Figure to Supplementary information. We hope you understand we kept Table 2 (Table 1 in the revised manuscript) because of its importance for comparing with other studies in the future.

Comment 3-2:

- For completeness, the labelling "left-handed" is missing in Fig.4b.

Reply 3-2:

Sorry, Figure 4b is removed by following another Reviewer's suggestion.

Comment 3-3:

- Figure 6 does not deliver too much information considering Fig.2, and the information contained in Fig.S8 is considerably important to be out. The authors should reevaluate this data and data-analysis presentation.

Reply 3-3:

We prepare a new Figure as shown in the following.

Figure 5. Analysis of CD and optical absorption spectra.

(a) Schematic diagrams of a symmetric band structure. (b) Schematic diagrams of an asymmetric band structure with a shift of Δ for 2_c and 2_v bands compared with (a). (c) Schematic CD spectrum of a (-)-enantiomer with the asymmetric band structure shown in (b). (d) CD spectra (upper panel) and absorption spectra (lower panel) of the sorted (-)-(6,5) SWCNTs. Four vertical dashed lines indicate the center positions of E_{12}/E_{21} , E_{13}/E_{31} , E_{24}/E_{42} , and E_{35}/E_{53} . The E_{ii} and E_{ij} transitions are depicted as purple and green strips, respectively. Individual single Lorentzian peaks were plotted using the energy values of all assigned transitions. The calculated CD spectrum (black dashed curve) is represented as the sum of all plotted peaks for comparing the experimental spectrum (red solid curve). Two unassigned peaks are labelled by green arrows.

Previous:

Figure 6 shows the CD spectra (upper panel) and absorption spectra (lower panel) of the M- and P-(6,5) SWCNTs separated in this work. Here, the CD and absorption spectra were normalized at the E_{22} peak. To check the purity of the (6,5) SWCNT enantiomers from the CD spectra, the $-1 \times$ M-enantiomer (denoted as -M) is plotted in Figure 6. Clearly, there is no difference between the P and -M spectra. This perfect coincidence indicated that our sample purity is high enough and we did not need to consider any impurities that might have affected the CD spectrum.

Revised:

Page 16, line 291:

Figure 5d shows the CD spectra (upper panel) and absorption spectra (lower panel) of the (-)-(6,5) SWCNTs separated in this work. Highly symmetrical CD spectra of (6,5) enantiomers (shown in Figure 2) indicated that our sample purity is high enough and we did not need to consider any impurities that might have affected the

CD spectrum.

Previous:

Using all assigned peak energies, the CD spectrum was reproduced by Lorentzian oscillator fitting. The calculated CD spectrum was highly consistent with the experimental spectrum (Supplementary Figure S8), except for four unassigned CD peaks at 2.54 eV, 3.14 eV, 3.78 eV and 4.75 eV.

Revised:

Page 17, line 302:

Although the transitions higher than E_{44} can be observed in high energy region, it is difficult to accurately assign them due to the complicated CD peaks. Using all assigned peak energies, the CD spectrum was reproduced by Lorentzian oscillator fitting (black solid curves). The calculated CD spectrum (black dashed curve) was highly consistent with the experimental spectrum (red solid curve), except for two unassigned CD peaks at 2.54 eV, 3.14 eV.

Comment 3-4:

- Fig.7 is interesting, but again, at that point of the development of SWCNT photophysics, it makes no sense to compare their optical transitions with DFT calculations. This figure could be in the supplementary information as a curiosity.

Reply 3-4:

We have moved this Figure into Supplementary information.

Comment 3-5:

- Fig.8 should be better explored to generate an empirical formula for E_{ii} and E_{ij} that should show consistency with previous works, such as the review [Araujo et al. Physica E 42, 1251-1261 (2010)].

Reply 3-5:

Based on the further analysis of the scaling behavior for obtained E_{ii} and E_{ij} transitions, we generate an empirical formula (shown in below), which is suitable for E_{ij} transitions as expected by Reviewer.

$$E_{ij}(p_{ij}, d_t) - \beta_{p_{ij}} \cos 3\theta / d_t^2 = a \frac{p_{ij}}{d_t} \left[1 + b \log \frac{c}{p_{ij} / d_t} \right]$$

Response to Reviewer #3:

This is definitely nice work describing the separation and CD spectroscopy of SWCNTs. The authors used the CD spectra together with selection rules to determine excitonic band structures of SWCNTs. I am not absolutely convinced that the work is suitable for Nature Communications. It would probably be better suited for a more specialized journal. In any case there are some points that need to be addressed before publication, as outlined below.

Reply:

We thank you for reviewing our manuscript and giving careful assessment and valuable recommendations on

our manuscript.

I believe that the paper reports the most complete and highest purity of chiral enantioseparated SWCNTs samples. However, the methods were reported before so the preparation of the samples is "just" an extension of previous work.

Reply:

We are very glad for receiving your high degree of recognition on our separation results of single chirality SWCNT enantiomers. In this work, however, we have highly improved the SWCNT enantiomer separation method using gel chromatography from our previous work. Differences between this study and our previous work are described below:

1. **Mixed-Surfactant.** High-resolution single chirality enantiomeric separation in this study was achieved by a mixed-surfactant composed of SDS, SC, and DOC. Only SDS surfactant was used in our previous work, which was not sufficient for high-purity separation.
2. **Stepwise elution.** The stepwise elution procedure (the DOC concentration was increased stepwise for fixed concentration of SDS and SC) was used to selectively elute the adsorbed individual enantiomers in this study. Only high concentration of SDS solution was used to elute all adsorbed SWCNTs without any selectivity in our previous work, which caused contamination of the other chirality.

As a result, we realized a large improvement in CD intensity for sorted enantiomers (see Table 1). Therefore, we claim that our method is "a newly developed method". To clarify this, we added a sentence as follows.

Previous: (none)

Revised:

Page 6, line 101:

Compared with our previous work¹⁷, much higher-resolution of single chirality enantiomeric separation was achieved in this study, which is mainly contributed by the newly introduced mixed-surfactant system and the stepwise elution procedure.

Specific comments:

Comment 1:

-There is no doubt that the samples are very pure, most of them are of higher purity than previously reported. However there is a weak point here that needs to be addressed / discussed. The authors quantify the purity of (n,m) species from the integrated intensity ratio of the absorption peaks. The analysis described assumes that the absorption cross sections are identical for all (n,m) species, which is not the case (Nano Lett. 2014, 14, 1530–1536).

Reply 1:

Basically, we agree with your opinion. The chirality-dependent absorption cross sections of (n,m) species should be considered for accurate purity evaluation. However, only a limited number of (n,m) species were reported in the literature you suggested. Therefore, we used a theoretical formula of oscillator strength reported in another literature (Nano Lett. **13**, 54–58 (2013)). We confirmed that this theoretical formula can well reproduce the diameter-dependent absorption cross section of SWCNTs. The corrected purity showed small deviations within a range of $\pm 1\%$ due to the small amount of impurities. To clarify this point, we have

revised the manuscript as follows.

Previous:

The purity of each (n,m) species was calculated as an integrated intensity ratio of the absorption peak of the target chirality to the sum of all other peaks using PeakFit software (Supplementary Figure S5). These results are shown in Table 1.

Revised:

Page 6, line 112:

The purity of each (n,m) species was **basically** calculated as an integrated intensity ratio of the absorption peak of the target chirality to the sum of all other peaks using PeakFit software (Supplementary Figure 5). **Since the absorption cross section of each (n,m) species is different^{23,24}, we have corrected the above purity using a theoretical formula of chirality-dependent oscillator strengths reported in ref 23. The final purity of each species is indicated in Figure 1a.**

Comment 2:

-Similarly, the authors use a normalized CD signal (CDnorm) to assess the enantiomeric purity of the samples. This is fine, but why do the authors use such a strange measure. Why not use the anisotropy factor $\Delta A/A$ (which is related to CDnorm), which is commonly used in CD spectroscopy?

Reply 2:

Thank you for your useful comment. It is true that the anisotropy factor g is commonly used for CD analysis. However, in the case of SWCNT, we cannot define the molecular weight of SWCNT. Therefore CD_{norm} has been commonly used in the previous works instead of anisotropy factor. In this work, we have to use this value to compare our value to others in the previous works. To clarify this, we revised the manuscript as follows.

Previous:

The relative enantiomeric purity of sorted enantiomers can be evaluated by a normalized CD intensity $(CD_{norm})^{13,14}$.

Revised:

Page 9, line 163:

The relative enantiomeric purity of sorted enantiomers can be evaluated by a normalized CD intensity $(CD_{norm})^{14,15}$, **instead of commonly used anisotropic factor that cannot be defined for SWCNT.**

Comment 3:

-The authors use P and M to refer to the enantiomers that showed plus or minus E11 CD signals. This is very misleading since P and M are reserved to describe the absolute configuration of molecules with helical chirality. The authors need to use other names!

Reply 3:

Thank you very much for valuable comment. We replaced the symbol P and M into (+) and (-), respectively, throughout the manuscript.

Comment 4:

-The authors speculate about the absolute configuration (handedness) of the SWCNTs on page 11 based on affinity arguments. This is very dangerous and highly speculative. The best thing to do would be to calculate the CD spectrum in order to assign the handedness. At least the authors should modify this part of the text.

Reply 4:

Thank you for your constructive comment. According to your suggestion, we added two authors and calculated CD spectra of (n,m) and $(n+m,-m)$ SWCNTs (enantiomers to each other) theoretically for (6,5) (Type 1) and (7,5) (Type 2). The results show that the CD signals of (n,m) and $(n+m,-m)$ are corresponding to that of $(-)$ - and $(+)$ - (n,m) SWCNTs in this work, respectively for Type 1 SWCNTs. Interestingly, as we speculated, the opposite result is obtained for Type 2 SWCNTs. We have also replaced the previous figure on the handedness of SWCNTs by the calculated CD spectra, and modified the corresponding discussion in the revised manuscript as follows.

Figure 4. Theoretical calculation of CD spectra for enantiomers.

(a,b) Calculated CD signal ΔW of enantiomers for (a) (6,5) (red) and (11,-5) (blue) and (b) (7,5) (red) and (12,-5) (blue), as a function of wavelength of incident light. Shaded lines correspond to E_{ij} and E_{ij} van Hove

singularity of joint density of states. (c) Reciprocal space of the graphene and cutting lines corresponding to Type 1 (left) and Type 2 (right) SWCNTs.

Previous:

However, our results showed that the separation order of M- and P-enantiomers depended on the type of SWCNT index. Our measurement could not determine which enantiomer was right-handed; thus, we might assume that right-handed SWCNTs had lower affinities for the gel and eluted at lower DOC concentrations. Based on this assumption, P-enantiomers could be identified as right-handed SWCNTs for Type 1 SWCNTs and left-handed SWCNTs for Type 2 SWCNTs, and vice versa. More simply, the CD signals of Type 1 and Type 2 SWCNTs have opposite signs. This phenomenon can be understood as a difference in the cutting lines for Type 1 and Type 2 around the K point in the Brillouin zone.

Revised:

Page 11, line 192:

However, our results showed that the separation order of (-)- and (+)-enantiomers depended on the type of SWCNT index. A possible explanation is that the CD signals of Type 1 and Type 2 SWCNTs with a specific handedness have opposite signs. This statement is supported by theoretical calculation of CD spectra of SWCNTs described below.

Previous:

Figure 4b illustrates the CD signal at E_{22} of left-handed and right-handed SWCNTs for Type 1 and Type 2 SWCNTs. Our finding suggests that the previous definition, namely, that SWCNTs exhibiting a positive CD peak at E_{22} can be identified as left-handed enantiomers and denoted as M, is not always correct^{9,13,14}; instead, the identification of left-handed enantiomers depends on the type of the SWCNT. The finding is supported by the theoretical study of CD spectra of SWCNTs by first-principles calculation¹¹.

Revised:

Page 12, line 207:

In Figure 4a, we show the calculated CD signal ΔW as a function of wavelength of the incident circular polarized lights in the direction perpendicular to the nanotube axis, for (6,5) (red line) and (11,-5) (blue line) SWCNTs in which (n,m) and $(n+m,-m)$ SWCNTs are enantiomers to each other. In the numerical calculation, ΔW is defined by the difference of optical absorption probability of (6,5) (or (11,-5)) SWCNT for the left- and right-handed circular polarized lights within tight-binding method for optical transition of SWCNTs³². It is important to note that the optical absorption probability near the K point for the right-handed circular polarized light is identical to that near the K' point for the left-handed circular polarized light because of the time reversal symmetry between the K and K' points, if we adopt an approximation of the vertical optical transition in the k space. It means that we would get $\Delta W = 0$ if we adopted this approximation. Thus it is essential in the calculation of ΔW to consider the *phase* difference of the light for different carbon atom positions that is usually neglected in the calculation of the absolute values of the optical absorption. The detail of the calculation will be reported elsewhere.

The calculated results reproduce the experimental results in many aspects: (1) ΔW of (6,5) SWCNT changes the sign at E_{22} , E_{33} , and E_{44} , and (2) ΔW of (11,-5) SWCNTs gives the opposite sign to that of (6,5) for all the wavelength of the light. Further, when we change (n,m) from (6,5) (Type 1 SWCNT) to (7,5) (Type 2 SWCNT) as shown in Figure 4b, we get the opposite sign of ΔW to that for (6,5) for each E_{ii} energy. This

phenomenon can be explained by the difference in the reciprocal space of the graphene and cutting lines between Type 1 and Type 2 SWCNTs as shown in Figure 4c. The optical absorption for E_{ii} van Hove singularity of Type 1 SWCNT occurs at the opposite k points measured from the K point compared to that of Type 2 SWCNT, whose transition optical dipole moment around the K point gives the opposite sign³³. The opposite behavior between type 1 and 2 SWCNTs in the calculation is also consistent with the experimental expectation based on their type-dependent affinities with gel. Thus alternative behavior of ΔW as a function of the wavelength comes from the change of the wavefunctions on one-dimensional cutting line near the K point^{34,35}.

It should be mentioned that we measure CD of SWCNTs with random orientation in the experiment. For incident circular polarized light, there are two inequivalent geometries of SWCNTs: The propagating directions of the light are (1) perpendicular to the nanotube axis (the case of Figure 4a and 4b) and (2) parallel to the nanotube axis (not shown in this paper). Even though we should integrate the CD for possible direction of the light, we did not show this for simplicity. We can imagine that the case of (2) gives a minor contribution after integration of azimuth angle of spherical coordinate. Further, we did not consider the effect of the exciton in this calculation in the first approximation. Since the wavefunction of an exciton is given by linear combination of the Bloch functions along one-dimensional Brillouin zone (or cutting line)^{36,37}, we can say that the behaviors of CD do not change even after we consider the exciton effect. Thus the present calculated results are reasonable to simulate the experimental results.

Reviewers' Comments:

Reviewer #1 (Remarks to the Author)

The authors addressed all my previous critics, I recommend the publication of the manuscript.

Reviewer #2 (Remarks to the Author)

The authors revised their work based on the criticisms and comments I presented in my previous reports. I think the paper improved significantly and I have no further concerns. I now recommend publication in Nature Communications.

Reviewer #3 (Remarks to the Author)